# OpenReview forum: "From Model to Breach: Towards Actionable LLM-Generated Vulnerabilities Reporting"
_ICLR.cc/2026/Conference — Submitted to ICLR 2026_

### Official Review · Reviewer_Lv5j · 2025-10-30

**Soundness:** 3
**Presentation:** 3
**Contribution:** 2
**Rating:** 2
**Confidence:** 4

**Summary:**

The paper discusses the security of code generated by LLMs, noting that open-weight models still produce vulnerable code in scenarios that have been documented for years. To improve the evaluation and mitigation of these risks, the paper introduces two new metrics: Prompt Exposure (PE) and Model Exposure (ME). The PE score assesses the severity of a vulnerability generated in response to a specific prompt by accounting for the vulnerability's inherent severity. The ME score then provides an overall summary of a model's security based on an aggregation of its PE scores.

**Strengths:**

- The paper demonstrates that even the latest open-weight models are vulnerable in the earliest reported vulnerability scenarios in a realistic use setting. Even the best models generate between 10% and 40% of code snippets that are vulnerable in well-documented and widely known scenarios
- The authors create an automated benchmark for security evaluation by extending the AATK dataset

**Weaknesses:**

- While the paper states they are the first to propose a systematic approach for analyzing LLM-generated code vulnerabilities, the core empirical foundation of the evaluation relies on existing work (Asleep at the Keyboard Paper)
- The methodology relies entirely on CodeQL to detect vulnerabilities in the generated code. If a prompt reformulation is not precise enough, the vulnerability patterns expected by CodeQL may not be present in the generated code.
- The definitions of Prompt Exposure (PE) and Model Exposure (ME) rely on specific assumptions about the Common Vulnerability Scoring System (CVSS) and is limited to this system

**Questions:**

Did you also check for false positives of the static analysis?

---

> ### Author Response · Authors · 2025-12-04
>
> We thank the reviewer for a thorough review and for positive feedback on the paper's presentation.
>
> As we mentioned to other reviewers, the use of Python subset of AatK is an explicit choice to demonstrate the lack of security improvement in the code generating LLMs since the AatK paper was first published 4 years ago. AatK is one of the earliest and most visible reports of code vulnerabilities generated by coding LLMs, and due to its prevalence and proximity to the English language, Python has historically been the domain where coding LLMs performed the best. We would hence have expected the Python vulnerabilities reported in AatK to have been included in post-training for coding LLMs and thus mitigated, but - surprisingly - this is not what we see.
>
> W1: We hope that above answers the reviewer's concerns.
>
> W2: We thank the reviewer for the remark. While we experimented with different vulnerabilities detectors and obtained similar results (Q3, reviewer TwVN), we did not report them.
>
> W3: We thank the reviewer for the remark and refer to our response to the reviewer TwVN (W4)
>
> Q1: We thank the reviewer for the suggestion; we were not aware of this possibility and the need to test for false positives

---

### Official Review · Reviewer_TwVN · 2025-10-30

**Soundness:** 2
**Presentation:** 4
**Contribution:** 2
**Rating:** 2
**Confidence:** 4

**Summary:**

This paper proposes two new metrics, namely Prompt Exposure (PE) and Model Exposure (ME), for quantifying the severity of vulnerabilities generated by LLMs in response to a prompt and grouped over multiple prompts for a model respectively. To do this, the Asleep At The Keyboard (AATK) benchmark is filtered for Python snippets with vulnerabilities that can automatically be detected by CodeQL -- resulting in a benchmark with 17 snippets. The Prompt Exposure (PE) for a prompt prone to a CWE (vulnerability type) is computed as follows: first the severity score for the CWE is approximated using the CVSS-B scores for associated CVEs. Further, the probability of generating vulnerable completions with the prompt is computed by sampling completions and evaluating them with CodeQL, and the likelihood of the prompt being used is also computed. The PE is then computed using these three values. The Model Exposure (ME) is computed by aggregating the PE scores for various prompts for a model.
On the AATK benchmark, the paper finds that the ME scores are more useful in ordering LLMs by the severity of vulnerabilities and this ordering differs from that induced by only looking at percentage of vulnerable completions by the model. Other findings include the sensitivity of certain prompts to reformulations and certain models being more prone to vulnerable code completion with prompt changes.

**Strengths:**

- The idea of severity scores for LLM generated code is novel and timely. Moreover, the consideration of differences in prompts and vulnerability types makes the metrics more fine-grained and compelling to use.

- The evaluation of newer LLMs on the not-so-new AATK benchmark is also very useful in pointing out that secure code completion should still be an active area of research. Further, the proposed ME scores induce an ordering over LLMs taking the severity of completions into account, which should be useful in practice.

- The paper was written well and the key ideas, specifically about how the metrics are computed, were easy to read.

Overall, I think this is an important problem to work on and the proposed metrics considering prompt and CWE variance is novel and seems useful.

**Weaknesses:**

- The title mentions "actionable" reporting. Which the ME scores can be useful in choosing between LLMs, I find it difficult to understand how these scores guide actions / future research for specific models. The prompt-level granularity offered by PE scores might be too specific to take any meaningful actions for updating these models. A CWE-level metric, not explored in this work, would be more "actionable" in the sense that it could identify specific vulnerabilities that any model is prone to generate.
- The value of the proposed metrics is tied to a benchmark that can support them. The benchmark here is arguably quite small (17 Python snippets) to generalize the findings to more samples from other languages, vulnerability types.
- Since one of the key ideas for defining the Prompt Exposure metric as such relies on including the probability of vulnerable generations and likelihood of prompts, I would have expected a discussion / experiment on how ablating these components affects the score. The components would be deemed more useful only if the ordering induced by them differs significantly from that just induced by raw CVSS scores.
- While the paper notes this limitation, I would have expected a more detailed justification for why it is okay to approximate the score for a CWE using CVEs tagged with it (since a given CVE might be tagged with multiple CWEs). The severity of CWEs is also expected to be language specific (i.e., some languages are more prone to certain vulnerabilities over others). At the very least, I would expect the sensitivity of the score to the CVEs used in the computation to be discussed (i.e., if I changed the cut-off date or randomly sampled some CVEs corresponding to a CWE, how much does the score change?).

Overall, I think that the paper could benefit from additional discussion and / or experiments justifying the various components of the metric and how it can be extended beyond the 17 Python snippets considered here. Further, a discussion on how this can benefit future work would be very helpful.

**Questions:**

I will summarize my questions from the weaknesses section above (please refer to that section for more details):

1. How do the proposed metrics promote action / guide future work?
2. How sensitive is the PE score to the choice of the set of CVEs used (either by cut-off date or random selection)?
3. How does the ordering change if various components of the score are ablated?
4. How well do you think these results would generalize to benchmarks from other languages and other vulnerability types?

---

> ### Author Response · Authors · 2025-12-04
>
> We thank the reviewer for a thorough review and for positive feedback on the paper's scope and presentation.
>
> While we agree with the reviewer that ME/PE metric is a major contribution of our method, we believe a much more critical and time-sensitive aspect of our contribution is the demonstration that we do not observe the improvement in generated code safety that most users expect, given the progress in other aspects of the generated code.
>
> From this perspective, using a Python subset of the Asleep at the Keyboard dataset is particularly relevant and, in our opinion, strengthens our point. AatK is one of the earliest and most visible reports of code vulnerabilities generated by coding LLMs, and due to its prevalence and proximity to the English language, Python has historically been the domain where coding LLMs performed the best. We would hence have expected the Python vulnerabilities reported in AatK to have been included in post-training for coding LLMs and thus mitigated, but - surprisingly - this is not what we see.
>
> W1: This is an excellent question. We designed the PE metric as a pendant to the CVSS scores used in cybersecurity to prioritize vulnerability patching. As such, it does not correspond to a CWE, but a specific instance of a CWE corresponding to a prompt-model-generated vulnerability, potentially affecting several models, so it is closer to a "CVE". As such, the "CWEs" in the figures should be more correctly identified as "AatK-Python-CWE-22" rather than "CWE-22". We have revised the paper to better explain this aspect of PE and its intended use.
>
> W2: We agree with the reviewer that demonstrating the value of ME/PE metrics would require significantly larger datasets, such as those put forward by the reviewer hkbs, and plan to revise the paper to include them
>
> W3: This is an excellent suggestion. The exploration of ME/PE stability with respect to sampled CVSS scores and the ability of perplexity normalization to suppress the impact of unnatural jailbreaking prompts that would not be expected to be seen in the real world is essential for demonstrating the value of ME/PE as vulnerability metrics.
>
> W4: This is an excellent remark. While we did perform CWE-related CVSS re-sampling at different points in time and observed consistency in the ME/PE metrics, we did not report this in the submitted manuscript and will adjust it. Overall, the CVSS score itself has attracted significant criticism in the cybersecurity community, especially for isolated code snippets, given that substantial risk can be mitigated by restricting access to the code only to the attacker. There is a substantial amount of generated vulnerable code insertion threat modeling that needs to be performed before vulnerability injection by coding LLMs can be properly qualified for severity. To the best of our knowledge, ME/PE scores are a first attempt at such a characterization, using an approximation that, although imperfect, is consistent with CVSS 4.0 guidelines.
>
> Q1: We hope that ME, defined over a larger body of reported code-generation vulnerabilities (17 AatK prompts is illustrative), would become a benchmark metric against which released coding models are judged, in the same way they are now judged on other performance metrics, such as HumanEval+. Similarly, we see PE used to report the severity of newly discovered code-generation vulnerabilities to attract the attention of developers who might need to patch codebases that could have been affected by such vulnerabilities, especially high-impact, high-severity ones.
>
> Q2: PE score is minimally impacted by the choice of CVEs selected to represent a CWE. However, we do not present this data in the manuscript, and we thank the reviewer for highlighting the need to include it.
>
> Q3: We have run preliminary analysis of the CyberSecEval dataset with encouraging results, before discovering issues with it, which have been reported by other teams. Overall, we would expect it to generalize well, but now we need actually to demonstrate this. We thank the reviewer for drawing our attention to its importance.

---

### Official Review · Reviewer_PToF · 2025-11-01

**Soundness:** 2
**Presentation:** 2
**Contribution:** 1
**Rating:** 0
**Confidence:** 5

**Summary:**

This paper deals with vulnerabilities in LLM-generated code.
It takes inspiration from the "Asleep at the Keyboard" (AATK) paper from 2021, and builds on its dataset. The original AATK dataset contains 54 code completion scenarios that could generate a vulnerability in a specific CWE category. The new AATK dataset proposed is smaller with just 17 scenarios (only in Python). As for AATK, CodeQL is used to assess the security of the completed code.
The paper then proposes a derivitative of the FIRST Common Vulnerability Scoring System SIG (CVSS) metric to deal with snippets.
The paper concludes that coding LLMs still cannot be trusted to write secure code, and that small prompt variations can strongly affect security.

**Strengths:**

- Understanding the security of LLM-generated code is an important topic

**Weaknesses:**

- The novelty (of dataset and scoring method) is very limited
- The resulting dataset is smaller than the original, raising the question about the contribution.
- The paper writing is verbose and chaotic -- it is very hard to pinpoint the precise contributions and arguments. Some text, like section 3.1, contains no relevant information. Also section 4.1 for some reason discusses LLM performance on HumanEval, which seems unrelated.
- The findings are inconclusive and unsurprising. The conclusion that "coding LLMs still cannot be trusted to write secure code" appears to suggest that at some point they will, which unfortunately will never be the case.

**Questions:**

I have no further questions

---

> ### Author Response · Authors · 2025-12-03
>
> We thank the reviewer for their feedback.
>
> To clarify, we do not believe that the dataset derived from the Asleep at the Keyboard paper is novel or interesting in itself. Rather, our main point is that we do not observe a security improvement in the latest generations of open-weight models that we would expect. We believe that using a subset of one of the earliest and most notable reports of vulnerabilities in LLM-generated code, in a language LLMs have historically performed well in, makes this point all the stronger.
>
> Addressing the reviewer's points specifically:
>
> W1-2: We do not claim dataset novelty. We believe that using a well-known set of prompt-vulnerability pairs allows us to make a stronger point about the lack of security improvements in the coding LLM space.
>
> W3: We thank the reviewer for identifying opportunities for the paper's writing improvement, although we are somewhat puzzled by the fact that other reviewers (TwVN, Lv5j) indicated the presentation as a strong point. As we mentioned in the paper, we used HumanEval to select older models that were sufficiently good at generating code to test for vulnerabilities, and compare their security with that of newer models.
>
> W4: While we are not willing to take a position on the reviewer's final claim, we believe that quantitatively demonstrating the current state of security and the lack of its improvement is more actionable for coding LLM developers and their users and can allow for future practices improvement, motivating the submission of this work.

---

### Official Review · Reviewer_hkbs · 2025-11-02

**Soundness:** 2
**Presentation:** 2
**Contribution:** 2
**Rating:** 2
**Confidence:** 3

**Summary:**

The authors propose a new evaluation framework for measuring how frequently LLMs generate insecure code and how severe the vulnerabilities are. The main contribution is the definition of Prompt Exposure (PE), which is a risk score that combines the severity of the vulnerability (via a representative CVSS score for each CWE class), the probability of generating vulnerable code, and how likely the prompt is to appear in real-world developer usage. This is aggregated across prompts to compute a Model Exposure (ME) score, enabling ranking of LLMs by security risk.
The authors test open-source models (comprising CodeLlama, Qwen, DeepSeekCoder, StarCoder, & Qwen variants) using 17 prompts adapted from Asleep at the Keyboard, with 10 paraphrases per prompt (generated by and then manually selected from GPT 3.5) and 25 model generations per paraphrase. They also evaluate human-level code correctness capability via HumanEval, Multilingual HumanEval, and HumanEval-Instruct. They show that modern models still frequently generate vulnerable code for many CWE categories, even when prompted in slightly different ways, and that any safety-tuning of more recent models has not eliminated these behaviours.

**Strengths:**

The novel risk-based metrics PE and ME, allowing models to be ranked by their security exposure.
Figure 2 is one of the strongest components of the paper. It clearly visualises the variance in vulnerability generation across CWE categories, models, and paraphrased prompts, demonstrating that some models consistently produce insecure code across minor prompt perturbations. However, this analysis is limited by the extremely small dataset of 17 prompts, all inherited from Asleep at the Keyboard.

**Weaknesses:**

* the work does not adequately position itself within the growing body of work on evaluating LLM-generated code security. Several benchmarks now exist that specifically assess vulnerability generation and, in many cases, also evaluate functional correctness, thus addressing exactly the functionality–security trade-off the paper discusses. These include Sec-Code-Bench, SecRepoBench, SafeGenBench, and benchmarks that jointly test correctness and security such as CWEval, SecCodePLT, BaxBench, and CodeGuard+. The absence of comparison or discussion of these efforts makes the contribution appear less novel and less grounded in current literature.
* the evaluation is limited to Python, which restricts the generality of the proposed PE/ME framework.
* the evaluation is limited only to open-source models. which is not customary for security papers.
* the evaluation is based on only 17 Python-based vulnerability-inducing prompts from Asleep at the Keyboard - and even with 10 paraphrases each, this is a very limited test set for drawing generalisable conclusions or ranking models by security exposure. This is particularly problematic given the existence of the much larger benchmarks which contain hundreds to thousands of vulnerable examples across diverse CWE categories.
* Although a “Valid” metric is reported for the AATK prompts, this only measures whether the generated code is syntactically valid (passes py_compile), not whether it produces correct output, or performs the intended task.

**Questions:**

•  Why do you only use 17 vulnerability prompts? Do you believe this is sufficient to rank models?
•  Why are existing security evaluation frameworks not used as baselines?
•  Do you check whether vulnerable code is functionally correct beyond Valid%? For example, does the code run and produce correct outputs?
•  Would your metric detect malicious prompt-induced vulnerabilities (jailbreaks), not just accidental ones?

---

> ### Author Response · Authors · 2025-12-03
>
> We thank the reviewer for their feedback and suggestions, particularly regarding the prior literature on code security benchmarking.
>
> While we agree with the reviewer that ME/PE metrics are a major contribution of our method, their proposal is motivated by the lack of improvement in the security of generated code that we observe. From this perspective, the Asleep at the Keyboard dataset is particularly interesting, as it is one of the earliest and most visible reports of code vulnerabilities generated by coding LLMs. Vulnerabilities reported in it would have been expected to be removed in the latest models, but this was not what we observed in the tested models. We apologize for not having featured this critical finding more prominently in the submitted manuscript.
>
> Addressing the reviewer's remarks specifically:
>
> W1: We agree with the reviewer that we need to properly cover prior research and position this work within it. While the prior work overview was accurate at the time it was written, it was not updated along with the rest of the paper prior to submission, and this is an oversight we addressed in the updated paper.
>
> W2: We believe that focusing on Python makes our point regarding the lack of generated code improvement stronger. Python's prevalence as a programming language and its proximity to English made it historically one of the languages on which LLMs performed better and an early focus for code generation benchmarks. We expect any code security improvements in coding LLMs to be more visible in Python than in other programming languages.
>
> W3: We understand the limitations of focusing on open-weight models, given that the models most used in practice by developers are API-only. Unfortunately, the code vulnerability studies' status is unclear with regard to EULA of such API-only models, and we were not granted an explicit permission to conduct this study by several such providers.
>
> W4: We agree with the reviewer that a proper demonstration of PE/ME would require an evaluation across several vulnerability datasets
>
> W5: We agree with the reviewer that the functional code evaluation is imperfect. It was added specifically for the AatK dataset to account for the lower vulnerability rate in non-functional code, and to normalize the ratios to the functioning fraction of code.
>
> Q1-2: We focus on a subset of AatK to demonstrate that even well-known vulnerabilities reported for almost 4 years at the time of the latest models' release are still not patched. We introduce ME/PE as ways to prioritize patching downstream, but we agree with the reviewer that these metrics require a more thorough benchmarking
>
> Q3: We agree with the reviewer that a functional evaluation would be essential. While we are most concerned with the vulnerabilities a developer might accidentally add to their codebase by copying generated code, it is unlikely that such code would be added without being functional.
>
> Q4: The malicious prompt-induced vulnerabilities are not part of the threat model considered here. Specifically, we focus on the vulnerabilities developers are likely to introduce in their code as part of their normal workflow, and it is unlikely that developers in this context would use malicious prompts. We specifically attempt to suppress adversarial prompts by normalizing by the prompt perplexity in the PE/ME definition.

---

### Meta-Review · Area_Chair_bJE9 · 2025-12-17

**Summary:**

This paper proposes two novel metrics, Prompt Exposure (PE) and Model Exposure (ME), to evaluate the security of code generated by LLMs. The authors demonstrate that even recent models remain vulnerable to well-established attack scenarios from the "Asleep at the Keyboard" (AATK) benchmark. While reviewers acknowledged the importance of the problem and found the core idea of the PE/ME metrics promising, a strong and unanimous consensus emerged regarding several critical weaknesses that lead to a recommendation for rejection.

The reviewers' concerns can be summarized in the following points:
  **Insufficient Evaluation:** The most significant concern, shared by all reviewers, is that the paper's conclusions are drawn from a very small and outdated dataset of only 17 Python prompts. This is not substantial enough to support generalizable claims or properly validate the new metrics.

  **Lack of Engagement with Prior Work:** The paper fails to engage with the extensive recent literature on LLM code security benchmarks. By not positioning the work within the context of existing, larger-scale evaluations, the paper's contribution and novelty are significantly diminished.

  **Limited Impact and Unsurprising Findings:** As a consequence of the limited evaluation and lack of comparison to modern benchmarks, the findings are perceived as unsurprising, and the paper's overall impact is limited.

  **Lack of Metric Validation:** Beyond the weak experimental setup, reviewers noted that the proposed PE/ME metrics themselves lack rigorous validation. The paper provides no ablation studies to justify their formulation's complexity, nor any analysis to test the robustness of their underlying assumptions.

**Reviewer Concerns:**

The authors' rebuttal acknowledged most of the reviewers' concerns but did not adequately address them in a way that would salvage the current submission.

**Outstanding Concerns:**
  **Limited Evaluation and Dataset (All Reviewers):** This remains the most critical issue. The authors' justification for using the small AATK dataset was to demonstrate that old vulnerabilities persist. While this explains their intent, it fails to address the reviewers' fundamental criticism that the dataset's scale is insufficient to empirically support the proposed quantitative metrics (PE/ME) or the resulting model rankings. The authors concede that proper validation would require a larger dataset, which leaves the paper's empirical foundation weak.

  **Lack of Engagement with Prior Work (hkbs, PToF):** The authors' admission that this was an "oversight" and that the related work section was not updated directly confirms the accuracy of the reviewers' criticism. The current manuscript is therefore deficient in its literature review and fails to properly contextualize its contribution.

  **Insufficient Validation of PE/ME Metrics (TwVN):** The rebuttal agrees on the importance of ablation studies and sensitivity analyses but provides no new analysis, instead relegating these to future work. As a result, the utility and superiority of the PE/ME metrics remain unproven in the current submission.

  **Absence of Functional Correctness Analysis (hkbs):** The authors also concurred that a functional evaluation of the vulnerable code is essential. However, this analysis is still absent from the paper, which weakens its assessment of real-world risk.

**Partially Addressed Concerns:**
  **Narrow Scope of Study:** The authors provided reasonable explanations for focusing only on Python and open-weight models, citing language prevalence and EULA constraints. While these justifications are understandable, they do not alter the fact, as noted by reviewers, that the study's scope is narrow. This limitation constrains the generalizability of its conclusions.

  The rebuttal largely confirms the reviewers' assessments by agreeing that more work is needed, effectively positioning the current submission as preliminary rather than a complete, well-validated contribution ready for publication.

**Reviewer Scores:**

Given the authors' rebuttal, which largely conceded the major weaknesses identified by the reviewers without providing new data or analysis to counter them, I do not believe any reviewer would have raised their score.


- **Reviewer hkbs (Score: 2):** Would likely remain at 2. The rebuttal confirmed the reviewer's primary concerns about the limited dataset and missing literature comparison.

- **Reviewer PToF (Score: 0):** Would certainly remain at 0. The reviewer's core issues with novelty and dataset size were not refuted.

- **Reviewer TwVN (Score: 2):** Would likely remain at 2. The author's response to the excellent questions about metric validation and actionability was to agree they were good points for future work, reinforcing the reviewer's assessment that the paper is currently incomplete.

- **Reviewer Lv5j (Score: 2):** Would likely remain at 2. The core weaknesses identified by the reviewer regarding reliance on AATK and CodeQL were not substantively addressed.

---

### Decision · Program_Chairs · 2026-01-26

Reject